# Recycling Nanoarchitectonics of Graphene Oxide from Carbon Fiber Reinforced Polymer by the Electrochemical Method

**DOI:** 10.3390/nano12203657

**Published:** 2022-10-18

**Authors:** Li Ling, Chao Wu, Feng Xing, Shazim Ali Memon, Hongfang Sun

**Affiliations:** 1Guangdong Provincial Key Laboratory of Durability for Marine Civil Engineering, College of Civil and Transportation Engineering, Shenzhen University, Shenzhen 518060, China; 2Department of Civil Engineering and Environmental Engineering, School of Engineering and Digital Sciences, Nazarbayev University, Nur-Sultan 010000, Kazakhstan

**Keywords:** GO, CFRP, recycling, electrochemical method, nano carbon onion

## Abstract

In this paper, an electrochemical method was proposed to recycle nanoarchitectonics of graphene oxide (GO) from carbon fiber reinforced polymer (CFRP). In the recycling process, NaCl solution with varied concentrations (3% and 10%) and tap water were used as electrolyte, while the impressed current density varied from 2.67 A/m^2^ to 20.63 A/m^2^. The results indicated that in NaCl electrolyte, the obtained nanoarchitectonics of GO contained a large amount of nano-carbon onions (NCO) produced by etching CFRP, while high purity GO was produced when tap water was used as electrolyte. The higher current density improved the production efficiency and resulted in a finer GO particle size. The proposed recycling method of GO is economical and simple to operate. It also provides an alternate approach to handle discarded CFRP.

## 1. Introduction

Carbon fiber reinforced polymer (CFRP) is a composite material that has been widely used in high-tech fields, such as aerospace, automobile, wind-turbines, civil engineering, robotics, and sport equipment, etc. due to its excellent mechanical, fatigue and corrosion resistance properties [1,2,3,4,5,6]. However, the increasing use of CFRP also leads to an increase in CFRP waste [7]. For example, the first generation of aircraft made with CFRP components served from 1972, and some have already come to their end of life [8,9]. A similar problem will be encountered in the next generation of composite aircraft [10], where each aircraft will produce more than 20 t of CFRP waste [11]. Therefore, the proper recycling of CFRP has received great attention. Currently, methods of disposing of CFRP waste mainly focus on recycling carbon fibers through chemical processes, incineration, and mechanical peeling [12,13,14,15,16], which generally require toxic chemicals, expensive facilities, or complicated processes. Later, researchers developed an electrochemical method to recover carbon fibers from CFRP [17,18,19]. The problem with this method is that the electrolyte becomes a new source of pollution. Therefore, it is necessary to treat the electrolyte properly after electrolysis, otherwise, it may pollute the environment. Fortunately, graphene oxide (GO) has been found in the electrolyte solution when electrochemical methods have been applied to reuse CFRP [17]. This has inspired us to recycle GO from CFRP by using an electrochemical method. Actually, GO as a certain nanoarchitectonics, has been widely used by providing functional groups as active sites to connect small organic molecules, polymers, and other functional groups [20,21,22,23,24]. It has been used in applications relating to pharmaceutical carriers [24,25,26], the biological environment [27,28], energy fields [29,30], and nanoelectronics and optoelectronics [31,32,33,34,35,36], etc.

In recent year, the preparation of GO by electrochemical methods has been realized by researchers. For example, Hudson investigated the anodic erosion of graphite electrodes by the generated oxygen and prepared functionalized colloidal GO [37]. Peckett found that the porous GO obtained by electrochemical method has (C–OH and C–O–C) chemical groups on the surface, which can be used for ion exchange purposes [38]. Zeng presented a green electrochemical method to obtain GO nanosheet-modified electrodes, which showed a very good electrochemical catalytic activity to ascorbic acid [39]. Generally, in these studies, graphite was used as the anode, while HBr, HCl, HNO_3_, or H_2_SO_4_, etc. solutions were used as the electrolyte. GO suspension was obtained after energizing the electrochemical system for a certain period of time. This method has the advantage that the current and voltage can be accurately controlled, so the sample preparation becomes easier, controllable, and repeatable. However, graphite anodes and chemical solvents electrolytes are relatively expensive and the processing cost is high. In addition, the GO prepared by existing electrochemical methods are all in the form of suspended solutions dispersed in the hazardous electrolyte, which is not friendly for the environment. In order to solve this problem, researchers tried to use double deionized water as electrolyte [37]. However, due to the very low conductivity of the electrolyte, the yield and/or the quality of the GO requires improvement.

Therefore, in this research, we attempted to recycle nanoarchitectonics of GO from CFRP through an electrochemical method. CFRP was used as an anode to reduce the cost of anode material, while NaCl solution and tap water were used as electrolyte to achieve clean production, which had not been previously reported. In this research, electrolyte species and the current applied were varied to improve the recycling efficiency and quality of GO. After the electrolysis process, the morphology and composition of GO were characterized. The suggested method is simple in operation, economical, and has relatively higher efficiency. In addition, all the materials required for the electrochemical process are readily and commercially available, thus, the recycling process would be cost-effective and is important for mass recycling. Finally, the entire electrochemical recycling process produces minor hazardous gases, which are easy to control and which can be recycled, making the whole electrochemical process harmless to the environment. Through the suggested method, improved yield and quality of GO can be obtained when compared with that produced by using double deionized water electrolyte. Thus, the proposed method broadens the approach for high-quality re-use of the discarded CFRP.

## 2. Experiments

In order to recycle GO from CFRP and assess the quality of the product, the following experiments were conducted:(1)Synthesis of GO using CFRP as raw materials through an electrochemical method;(2)Characterization of the recycled GO from the aspects of composition, functional groups, morphology, and particle size distribution. Through characterization, the quality of GO has been evaluated and the recycling mechanism has been speculated.

### 2.1. Materials

CFRP was used to recycle GO through the electrochemical method. The CFRP strips are made of multi-layer carbon fibers (Toray T700, Toray Composite Materials America, Inc. (CMA), Tacoma, WA, USA) having a volume fraction of 60%, and are covered by LAM-125/-226 epoxy (Pro-Set Inc, Bay City, MI, USA), as shown in Figure 1a,b. The details of chemical composition of epoxy used in CFRP are shown in Table 1, while the thickness of CFRP strips was approximately 2 mm, as shown in Figure 1c. For recycling purposes, the CFRP strip was divided into three regions, i.e., test region, protected region, and electrical connection region, as shown in Figure 1a. The test region was used for recycling, and the protected region was sealed by epoxy resin, while the electrical connection region was used to connect the CFRP to the electrode. The detailed geometric dimensions of CFRP are shown in Figure 1a.

### 2.2. Recycling Procedure

The electrochemical method was adopted to recycle GO from CFRP as shown in Figure 2. For this purpose, the CFRP strips were connected to the positive terminal of a direct current (DC) power source as anode, while the stainless steel strips were connected to the negative terminal of a DC power source as cathode. In this research, NaCl solution and tap water, respectively, were used as electrolyte. For the NaCl solution, the concentration of NaCl solution was 3% and 10%, respectively, and deionized water was used as a solvent. For the NaCl electrolyte, constant currents of 4 and 10 mA were used, which corresponded to current densities of 2.67 and 6.77 A/m^2^ (relative to the anode area), respectively. The concentration of NaCl solutions and current densities were chosen according to the findings of our previous research to recycle carbon fiber from CFRP [17], where the use of these parameters provided reasonable recycling efficiency and production quality.

For the tap water electrolyte, the constant currents of 4, 10, and 30 mA, respectively, were applied. The higher current of 30 mA was adopted here to improve recycling efficiency of GO, since the tap water has lower conductivity compared to NaCl solution. In both cases, after 21 days, the solution containing GO suspension was collected [17]. 

In the research, each specimen was labeled according to concentration of NaCl solution and impressed current. For example, the label I4S3 specimen indicates impressed current of 4 mA and NaCl solution with concentration of 3%. The detailed experimental matrix is presented in Table 2.

### 2.3. Testing Methods

The composition of GO was characterized through Raman spectroscopy technology using Renishaw InVia Reflex Microscope (Renishaw, Hoffman Estates, IL, USA) equipped with a 514 nm Argon laser. For this purpose, the samples were separated from the solution by using vacuum filtration with a pore size of 0.45 μm. Subsequently, the filter paper containing GO was air dried and used for Raman testing. 

The functional groups of GO were characterized by Fourier Transform Infrared Spectroscopy (FTIR) with a Spectrum One FTIR spectrometer (Perkin-Elmer, Waltham, MA, USA). The scanning range was 400–4000 cm^−1^ at 2 cm^−1^ resolution. Before the testing, 0.3 g of the air dried GO powder and 30 g of KBr powder were mixed and ground by hand under an infrared baking lamp. Then, 0.05 g of the ground powder was pressed into a transparent pellet for FTIR testing.

In order to further identify the composition of GO, ultraviolet and visible spectra (UV-VIS) was measured using Perkin-Elmer Lambda 750 UV-VIS-NIR spectrophotometer (Perkin Elmer, Waltham, MA, USA) at a range from 190 to 600 nm. The final spectra of GO were obtained by measuring the spectra of electrolyte (NaCl solution having 3% and 10% concentration, respectively, or tap water), which was removed from that of the GO suspension. 

The morphology of GO was observed by using Scanning Electron Microscope (SEM) and Transmission Electron Microscope (TEM) techniques. SEM was performed using a FEI quanta 250 (EFI, Hillsboro, OR, USA) with an accelerating voltage of 15 kV and a working distance of approximately 10 mm at a low vacuum mode. During the observation, BSE detector was used to observe the morphology of GO to avoid the influence of NaCl particles precipitated from the residual NaCl electrolyte. TEM was performed using a JEM-2100F Field Emission Electron Microscope (JEOL, Tokyo, Japan) with an accelerating voltage of 200 kv. For TEM observation, the sample was prepared by loading a drop of GO suspension onto a copper grid coated with an amorphous carbon film, fully evaporating the water at room temperature. 

The particle size distribution of the GO was measured by Dynamic Light Scattering technology on a DelsaMax CORE nanolaser (Beckman Coulter, Brea, CA, USA). The samples were analyzed using a multi-channel correlator with a test range of 17 to 2000 nm.

## 3. Results and Discussion

### 3.1. Recycling of GO with NaCl Solution as Electrolyte

#### 3.1.1. Macrograph of Recycled Suspensions

The suspensions recycled through electrochemical method with NaCl electrolyte are shown in Figure 3. It can be seen that the color of the suspension became darker as the NaCl electrolyte concentration increases from 3% to 10% at a certain current. Therefore, the color of recycled suspension was significantly affected by the concentration of NaCl electrolyte. A darker color of suspension usually indicates a higher concentration of GO in suspension [40]. Therefore, the S10 group with higher NaCl electrolyte concentration seems to produce more GO, exhibit higher recycling efficiency, and have a better recycling potential than the S3 group.

It can be observed that at a certain electrolyte concentration and with the increase in the applied current from 4 mA to 10 mA, the color of suspension became darker suggesting that the increase in current also improved the recycling potential of GO. It is pertinent to mention here that the suspension remained well-dispersed even after one year at room temperature without any agglomeration illustrating good compatibility of GO with aqueous solution.

To further investigate the role of current in the recycling process, CFRPs were immersed in NaCl solutions with concentrations of 3% and 10% for 21 days and no current was applied. The results after immersion are shown in Figure 4. It can be seen that the CFRP did not deteriorate in appearance, and no change in the color of the solution was observed. Therefore, simple immersion in the electrolyte will not easily decompose the carbon in CFRP without applying current, which means that electricity plays an important role in the recycling process.

#### 3.1.2. Composition

The composition of the recycled GO was characterized by Raman and UV spectroscopy. The Raman spectra of samples with different concentrations of NaCl electrolyte and applied current are shown in Figure 5. From the spectra, it can be clearly seen that the spectrum of the I10S3 sample is closest to that of typical GO, which presents D, G, and 2D peaks centered at 1350 cm^−1^, 1610 cm^−1^, and 2800 cm^−1^, respectively [41]. The peak at 1350 cm^−1^ is the D band of GO, which is considered to be related to the defects or disorder part of graphite [42,43]. The peak at 1580 cm^−1^ indicates the existence of perfectly crystallized graphite. Therefore, all the GO suspensions contain sp^2^ hybridization but with a certain degree of defects. All the samples showed very little difference in I_D_/I_G_ ratio, which means that the crystallization level of GO products do not change much [43,44]. Finally, the peak at 2800 cm^−1^ is the 2D band of GO, which signifies the layered structure in GO [45]. The 2D band was observed in all of the GO specimens with a lower intensity than the G band, indicating the existence of a multilayer structure in GO [46].

The functional groups of the recycled GO were analyzed through FTIR technique, as shown in Figure 6. In the spectra, peaks at 3430 cm^−1^, 1650 cm^−1^, 1384 cm^−1^, and 1068 cm^−1^ were found, which represent the stretching vibration peak of O-H in carboxyl groups (-COOH), the stretching vibration peak of carbonyl group (-C=O), in-plane bending vibration of hydroxyl groups (-OH) and symmetric stretching of carbonyl group (-C=O), respectively [47,48,49]. The results indicate that the recycled nanoarchitectonics of GO contained a large number of oxygen-containing functional groups. The peak at 1120 cm^−1^ turned higher in specimens with higher NaCl concentration, which might be due to the inductive effect of Cl^−^ [50,51]. Furthermore, it could be observed that the peak intensities of these oxygen-containing functional groups gradually increased with the current, which might indicate that the oxygen-containing functional groups originated from the oxidizing of C-C bond of the CFRP anode in electrochemical system.

In order to further determine the composition of GO, UV spectra were obtained and results are presented in Figure 7. From the figure, two absorbance peaks can be observed. The first peak was observed at a wavelength of 230 nm, which originated from π–π* transition of aromatic C=C bond in GO [52]. The second peak was observed at approximately 199 nm, which is considered to be from the defective carbon nanostructures with a higher oxidation degree [53,54]. For all of the four specimens, high absorbance intensity at approximately 199 nm and a much lower absorbance intensity at approximately 230 nm were observed, indicating that much more defective carbon nanostructures were produced than GO of 230 nm in NaCl electrolyte. Moreover, it was noticed that the yield of GO increased significantly as the applied current intensity increased from 4 mA to 10 mA.

#### 3.1.3. Microstructure

The microstructure of the recycled specimens was investigated with SEM. The BSE technique was used to exclude the influence of NaCl particles as much as possible, where NaCl appeared brighter than GO due to the larger average atomic number when it was viewed at the same height as GO. The results of GO suspensions obtained at a different concentration of NaCl electrolyte and applied current presented in Figure 8 show finer carbon nanostructures in all four specimens, which is consistent with the analysis obtained from UV spectra.

TEM was used to further characterize the microstructure of GO specimens obtained at low and higher magnifications. The results samples obtained with different concentration of NaCl electrolyte and applied current at lower and higher magnifications are presented in Figure 9. It can clearly be seen that at a lower magnification, the edges of individual flakes can be distinguished. All specimens consist of two structures, i.e., layered GO (multi-layer) and spherical particles. The layers are smoothly stacked without ripples and wrinkles, which is similar to GO synthesized using atmospheric plasma, while spherical particles are attached to layered GO [55]. Under high-magnification, the spherical particles are in the shape of hollow carbon onions (NCOs) with a particle size of approximately 10 nm, similar to the defective carbon nanostructure in Ref. [54].

#### 3.1.4. Particle Size Distribution

To identify the particle size distribution of the GO suspensions obtained with different concentrations of NaCl electrolyte and applied current, a laser nanometer size analyzer was used. The results are shown in Figure 10. According to the results, it can be seen that the recycled suspensions have different particle size distribution in the order of I4S10 > I10S10 > I4S3 > I10S3. These indicate that the particle size of the suspensions recycled by the electrochemical method decreased as the applied current increased from 4 mA to 10 mA at a specific NaCl electrolyte concentration. Similarly, at a constant level of applied current, the particle size of suspensions increased as the NaCl electrolyte concentration increased from 3% to 10%. Therefore, higher current density and lower NaCl concentration in the solution decreased the particle size distribution of recycled suspensions. However, it should also be noted that the size of NCO (10 nm) was not precisely reflected by this method, possibly due to the agglomeration of NCO particles.

### 3.2. Recycling of GO with Tap Water as Electrolyte

#### 3.2.1. Macrograph of Recycled Suspensions 

The GO suspensions recycled through electrochemical method using tap water as electrolyte are shown in Figure 11. It can be seen that the color of suspensions became darker with the increase of applied current from 4 mA to 30 mA, indicating that the concentration of recycled suspensions can increase with the applied current. It can also be seen that all of the suspensions were well-dispersed even after one year of keeping the sample at room temperature, indicating a good compatibility of GO suspension with water.

#### 3.2.2. Composition

Raman tests were conducted to identify the composition of the GO specimens recycled through the electrochemical method with tap water as electrolyte. The Raman spectra of samples with different applied current are presented in Figure 12. It can be seen that the GO recycled with tap water has a similar composition to that obtained when NaCl was used as electrolyte, e.g., multilayer of GO with sp^2^ hybridization and a certain amount of defects. 

The functional groups of the recycled GO with tap water as electrolyte were also analyzed through FTIR technique, as shown in Figure 13. Similar peaks at 3430 cm^−1^, 1650 cm^−1^, 1384 cm^−1^, and 1068 cm^−1^ were found, with their positions and intensity trends similar to the previous results, as shown in Figure 6. It indicates that the carboxyl groups (-COOH), carbonyl groups (-C=O), and hydroxyl groups (-OH) were formed in the GO in the electrochemical system. The intensity of these peaks gradually increased with the current, indicating that the increase of current could significantly promote the accumulation of oxygen-containing functional groups. 

However, according to the UV spectra (Figure 14), the suspensions recycled from tap water showed significant differences in comparison to those recycled through NaCl as electrolyte. All suspensions recycled with tap water showed a main absorbance peak at 230 nm, which is similar to GO samples synthesized by using Hummers method [56]. It can also be seen that the intensity of spectrum increases with the increase of current from 4 mA to 30 mA, indicating that the concentration of GO significantly increases with the applied current. Therefore, the increase of current improves the production efficiency of GO.

#### 3.2.3. Microstructure

The SEM images of recycled suspensions obtained at different applied currents are shown in Figure 15a–c. The microstructures of suspensions recycled with tap water as electrolyte were greatly changed compared to those recycled with NaCl electrolyte. Other than the agglomeration of finer carbon nanostructures on the top of the filter paper, the recycled suspensions seemed to hang over the supporting edges of holes in the filter paper, which in turn made the edges of holes thicker. The TEM image of specimen I4S0 presented in Figure 15d shows a typical multilayer GO structure as presented in Ref. [57], which is consistent with the results of UV spectra obtained in Figure 14. The morphology of other samples observed through TEM showed little difference, when compared with I4S0 sample and are not presented here.

#### 3.2.4. Particle Size Distribution

The particle size distribution of GO suspensions recycled under varied applied current are presented in Figure 16. It can clearly be seen that the particle size of recycled suspensions significantly decreased with the increase of current from 4 mA to 30 mA. As the current increased from 4 mA to 10 mA, the average particle size decreased from 560 nm to 320 nm while the increase in the applied current to 30 mA resulted in the decrease in the particle size to 20 nm. The results indicate that the increase of current is beneficial for the refinement and homogenization of GO particles.

### 3.3. Mechanism of Recycling GO

According to the characterization results, the recycling mechanism of the GO suspension by electrochemical method was investigated. The schematic is shown in Figure 17.

When the NaCl solution was used as electrolyte, the electrolysis of NaCl was dominant and that of H_2_O occurred in a small amount [58]. During this process, Cl^−^ and OH^−^ ions were produced. Under the driven of applied electric field, the negative Cl^−^ and OH^−^ ions moved towards the anode, where Cl^−^ was oxidized to Cl_2_ ($2{\mathrm{Cl}}^{-}-2e^{-}\to {\mathrm{Cl}}_{2}↑$), while a small amount of OH^−^ was oxidized to O_2_ ($4{\mathrm{OH}}^{-}-4e^{-}\to 2H_{2}O+O_{2}↑$). Both Cl_2_ and O_2_ etched the CFRP; resulted in the formation of NCO particles and was found to be dominant in the recycled specimens. The remaining small amount of OH^−^ was also responsible for extracting GO from CFRP. During the process, NCO and GO was formed through etching CFRP, part of the C-C bond was oxidized into functional groups, such as carboxyl groups (-COOH), carbonyl groups (-C=O), and hydroxyl groups (-OH).

When tap water was used as electrolyte, the H_2_O molecule was ionized into H^+^ and OH^−^ ions ($H_{2}O\to H^{+}+{\mathrm{OH}}^{-}$). Under the driven of applied electric field, negative OH^−^ ions moved toward the anode and pulled graphite layer from CFRP to form the dominant multi-layer GO [59]. The application of higher current improved the production efficiency and refinement of GO particles, which is consistent with the observation presented in Figure 14 and Figure 16.

## 4. Conclusions

In this research, GO was recycled using an electrochemical method, where NaCl solution and tap water were used as electrolytes. By characterizing GO suspensions obtained through NaCl as electrolyte, it was found that the composition of GO was a mixture of multilayer GO and NCO. Higher current densities and lower NaCl concentrations favored the development of mixtures having a smaller particle size. While in the tap water electrolyte, typical multilayer GO was produced with high purity. The increase of current was beneficial to improve the recycling efficiency and refined the GO particles. The mechanism of GO recycling was considered to be related to the pulling effect of OH^−^ on CFRP, while the formation of NCO may be linked to the etching of CFRP by Cl_2_ and O_2_.

In comparison to the existing methods, this recycling method is simple to operate and economical. The materials used are easy to acquire and can be steadily supplied. In addition, when tap water was used as electrolyte, the obtained GO suspension was of high quality and less impurity. Thus, it expands the application field of the recycled GO from CFRP via the electrochemical method.

It should be mentioned here that the GO suspension recycled by this method may contain a small amount of polymer particles, resulting from the corrosion of polymer in CFRP. However, these components may have little impact in some fields. For example, the recycled GO can be used as a nano-reinforcement in concrete in civil engineering. For the field applications, where high purity GO is required, the CFRP anode can be replaced with graphite rods etc. Conclusively, the research provides an alternative option for the complete recycling of CFRP. Moreover, the recycling method could contribute to the synthesis of GO nanoarchitonics by using other carbon-based materials as anode.

## Figures and Tables

**Figure 1 nanomaterials-12-03657-f001:**
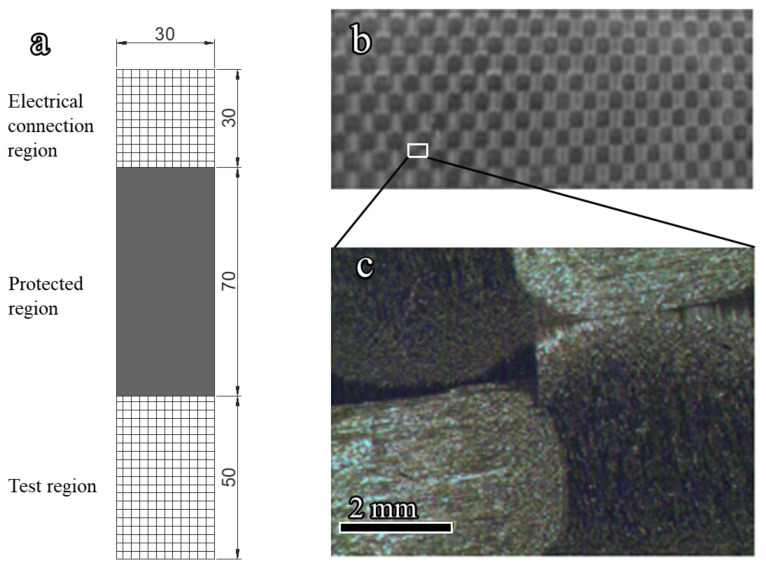
Geometric dimensions and the surface texture of CFRP specimen (unit: mm). (**a**) Geometric dimensions of CFRP specimen; (**b**) photograph of test region; (**c**) an enlarged view of the selected area of (**b**) taken by optical microscope.

**Figure 2 nanomaterials-12-03657-f002:**
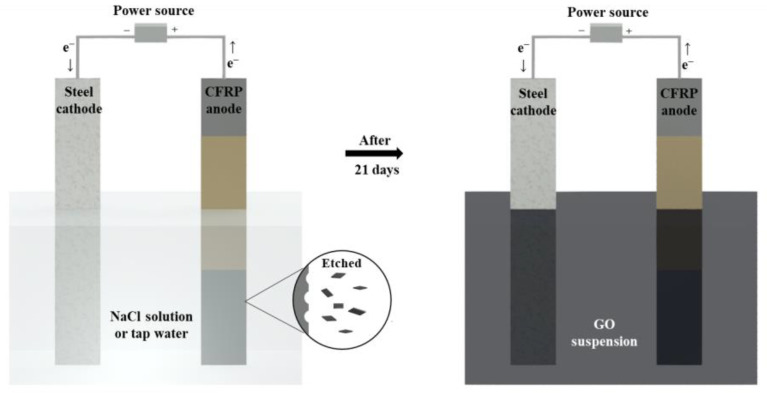
Scheme of recycling nanoarchitectonics of GO by electrochemical method.

**Figure 3 nanomaterials-12-03657-f003:**
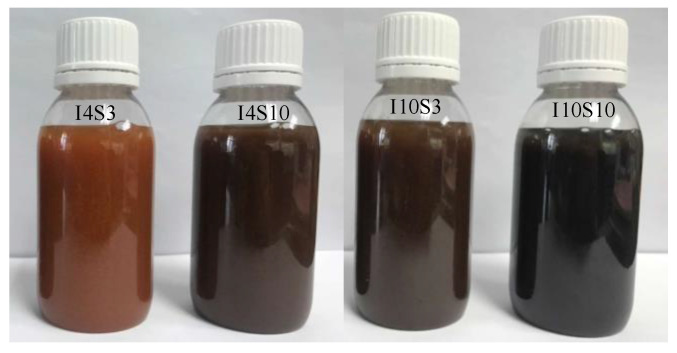
Suspensions recycled from NaCl electrolyte.

**Figure 4 nanomaterials-12-03657-f004:**
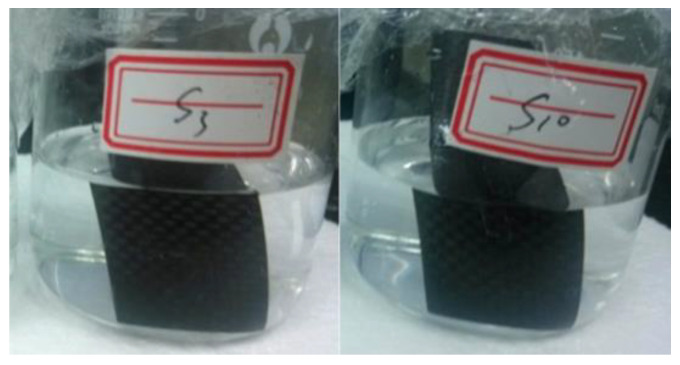
CFRP immersed in 3% (indicated by S3 in the image) and 10% (indicated by S10 in the image) NaCl solution for 21 days without applying current.

**Figure 5 nanomaterials-12-03657-f005:**
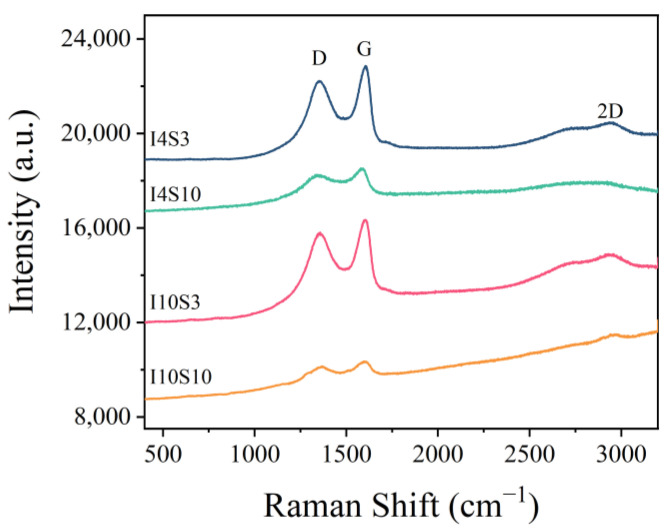
Raman spectra of specimens recycled from NaCl electrolyte.

**Figure 6 nanomaterials-12-03657-f006:**
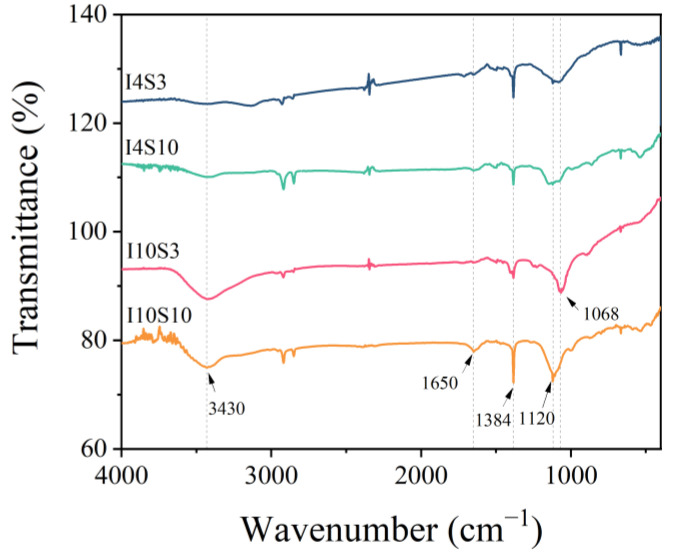
FTIR spectra of specimens recycled from NaCl electrolyte.

**Figure 7 nanomaterials-12-03657-f007:**
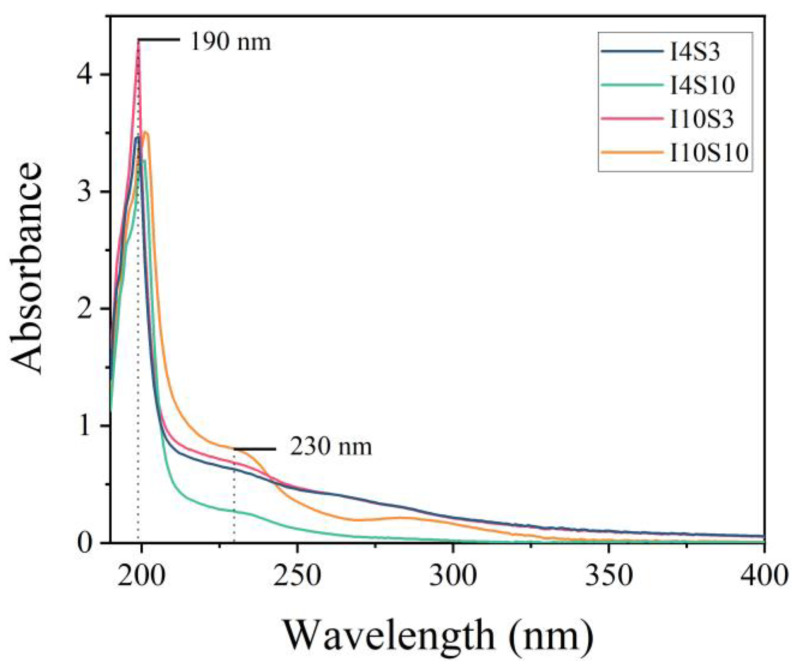
UV spectra of specimens recycled from NaCl electrolyte.

**Figure 8 nanomaterials-12-03657-f008:**
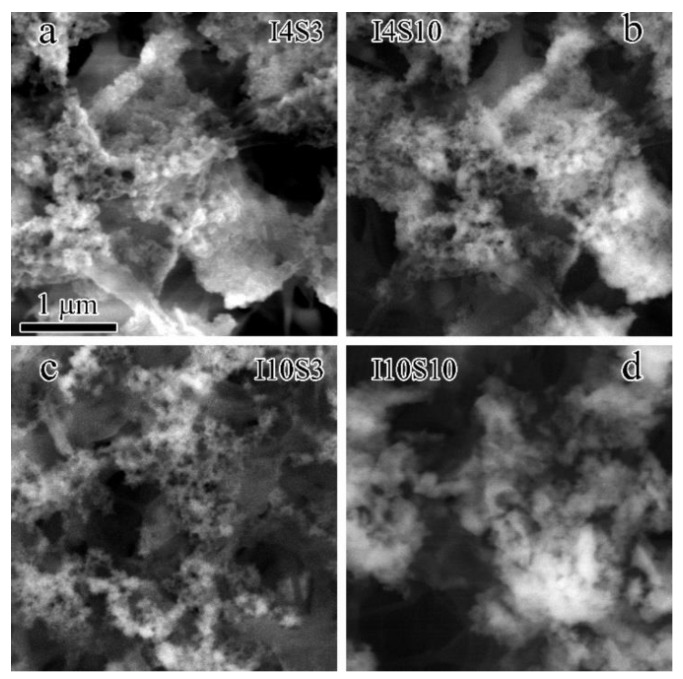
The SEM images of specimens recycled from NaCl electrolyte. (**a**) I4S3; (**b**) I4S10; (**c**) I10S3; (**d**) I10S10.

**Figure 9 nanomaterials-12-03657-f009:**
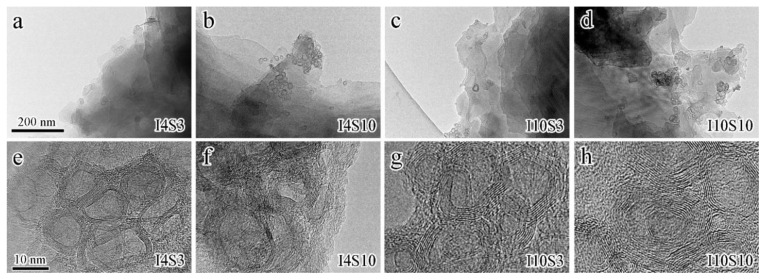
The TEM images of specimens recycled from NaCl electrolyte. (**a**–**d**), low magnification images of I4S3, I4S10, I10S3, and I10S10, respectively; (**e**–**h**), high magnification images of I4S3, I4S10, I10S3, and I10S10, respectively.

**Figure 10 nanomaterials-12-03657-f010:**
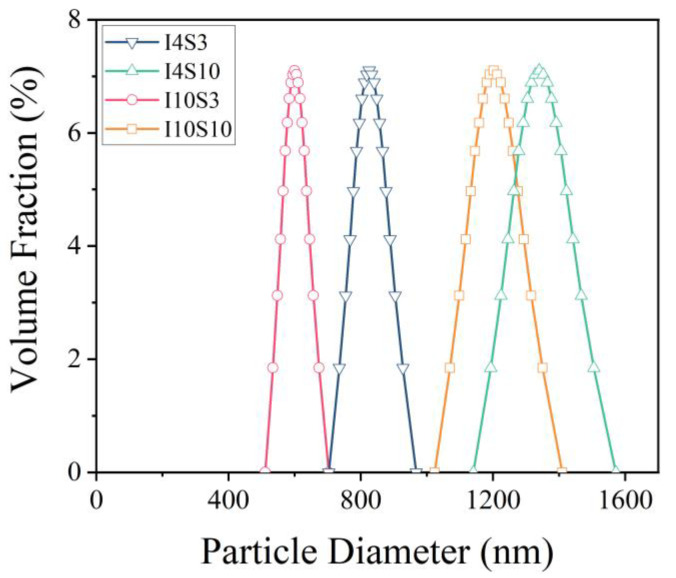
The particle size distribution of suspensions recycled from NaCl electrolyte.

**Figure 11 nanomaterials-12-03657-f011:**
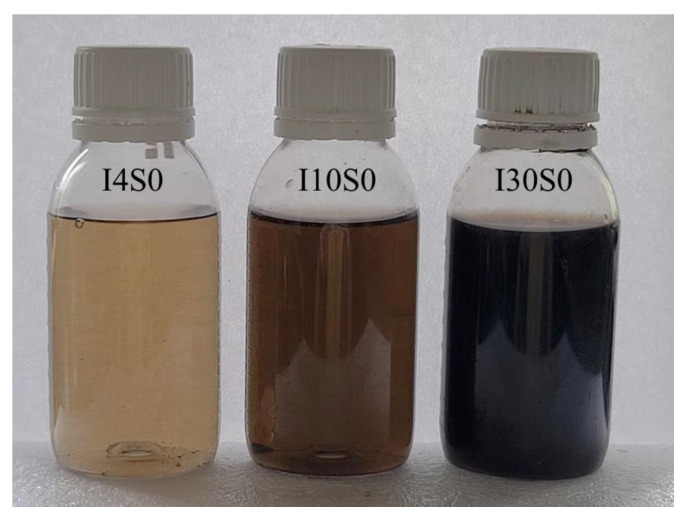
Suspensions recycled from tap water electrolyte (diluted by 5 times).

**Figure 12 nanomaterials-12-03657-f012:**
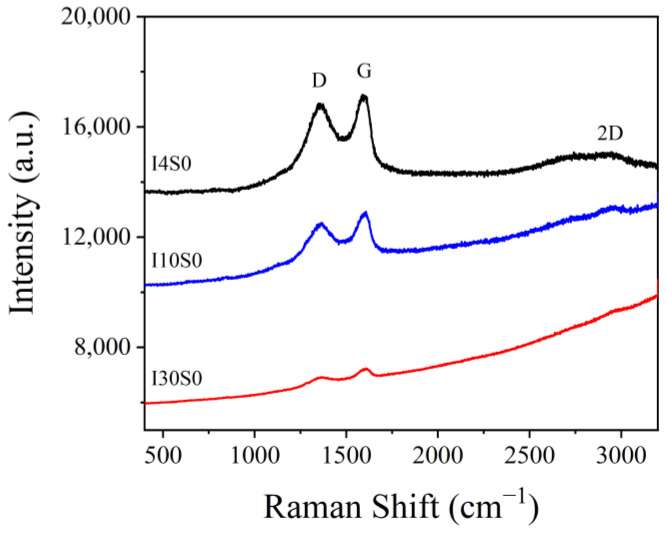
Raman spectra of specimens recycled from tap water electrolyte.

**Figure 13 nanomaterials-12-03657-f013:**
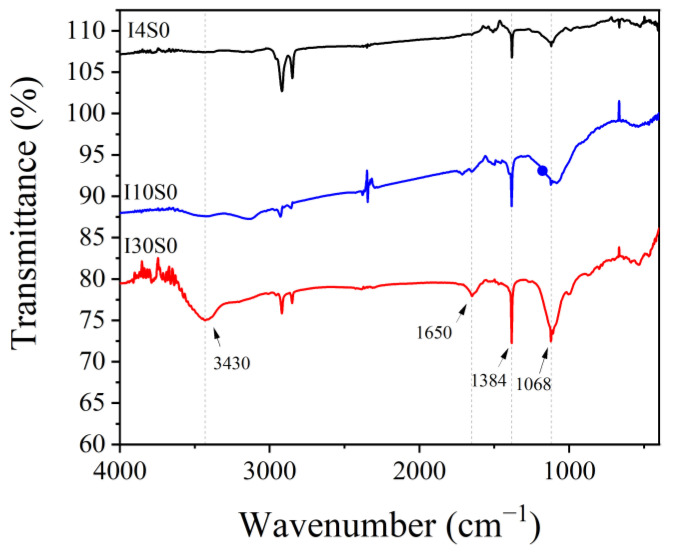
FTIR spectra of specimens recycled from tap water electrolyte.

**Figure 14 nanomaterials-12-03657-f014:**
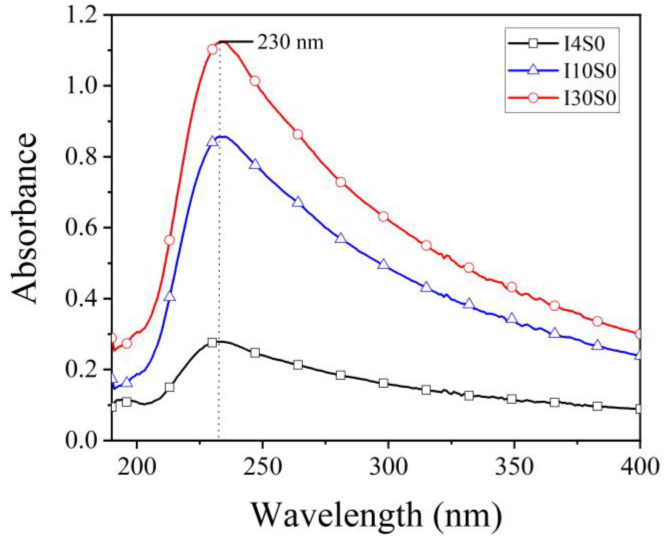
UV-VIS spectra of suspensions recycled from tap water electrolyte.

**Figure 15 nanomaterials-12-03657-f015:**
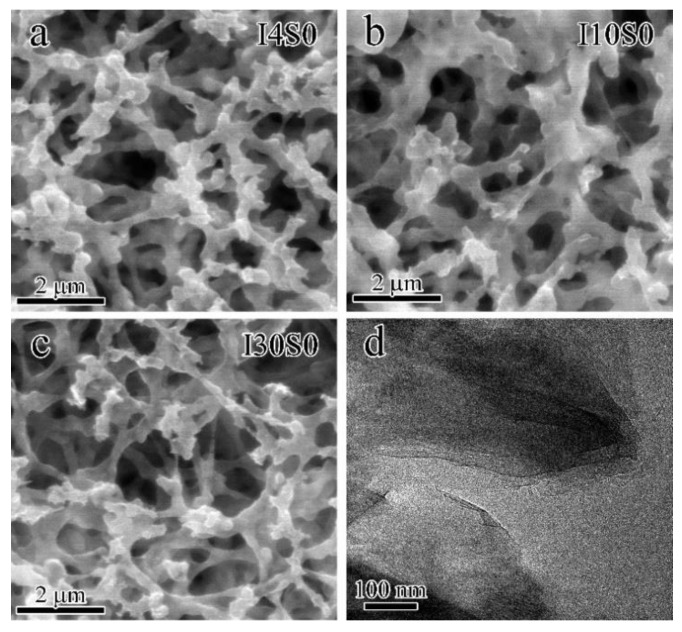
The SEM images ((**a**) I4S0; (**b**) I10S0; (**c**) I30S0) and TEM image (**d**) of specimens recycled from tap water electrolyte.

**Figure 16 nanomaterials-12-03657-f016:**
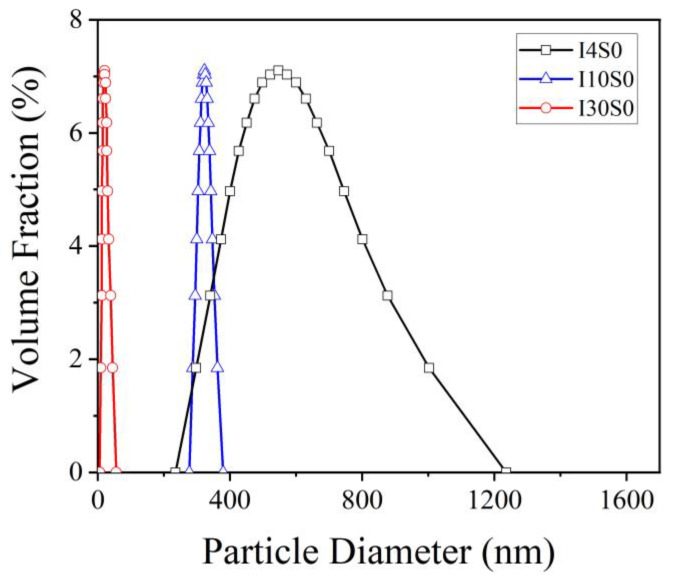
The particle size distribution of suspensions recycled from tap water electrolyte.

**Figure 17 nanomaterials-12-03657-f017:**
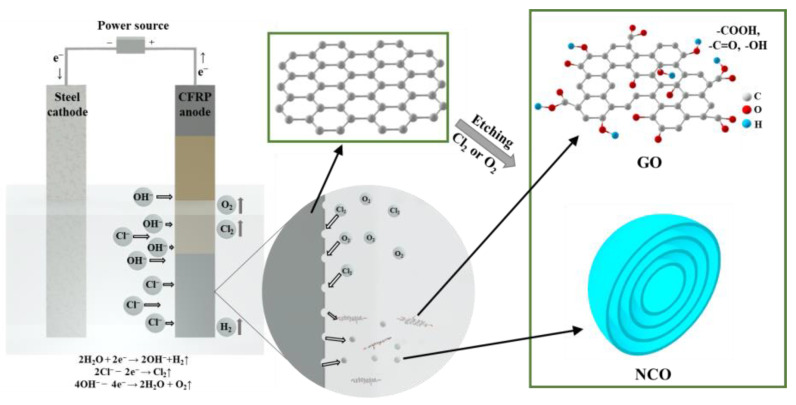
The schematic of GO recycling by electrochemical method.

**Table 1 nanomaterials-12-03657-t001:** Chemical composition of epoxy in CFRP.

Ingredient	Concentration (%)
Bisphenol-A type epoxy resin	37–38
Novolac epoxy resin	19–20
Dicyandiamide	5–6
Methyl ethyl ketone(MEK)	36–37

**Table 2 nanomaterials-12-03657-t002:** The details of experimental matrix.

Specimen	Current (mA)	Current Density (A/m^2^)	Concentration of NaCl (%)	Type of Electrolyte
I4S0	4	2.67	-	tap water
I10S0	10	6.77	-	tap water
I30S0	30	20.63	-	tap water
I4S3	4	2.67	3	NaCl solution
I10S3	10	6.77	3	NaCl solution
I4S10	4	2.67	10	NaCl solution
I10S10	10	6.77	10	NaCl solution

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
