# Peer review of "Recycling Nanoarchitectonics of Graphene Oxide from Carbon Fiber Reinforced Polymer by the Electrochemical Method"

_nanomaterials, 2022, doi:10.3390/nano12203657_

Round 1
Reviewer 1 Report
It is an practically important work. The data are somehow collected in convincing ways. I may recommend publication of this work in Nanomaterials. However, some revisions are necessary. Please see below.
1) In this work, recycling procedures are important. These processes have to be clearly visualized with appropriate figures and/or scheme. The current Figure 2 is simple cartoon of electrode systems. Please add one scheme of figure to explain detailed process and science of recycling.
2) Although this manuscript includes lots of important scientific and technological issues, the current title may give impression of simple engineering work. In such case, inclusion of conceptual term in the title is recommended. This is certain kind of nano-preparation with recycling process, and thus, I may propose use of conceptual term Recycling nanoarchitectonics (as post-nanotechnology concept, see https://pubs.rsc.org/en/content/articlelanding/2021/nh/d0nh00680g). For example, the title like … Recycling nanoarchitectonics of graphene oxide from carbon fiber reinforced polymer by electrochemical method … may sound more attractive.
3) The prepared materials are structurally analyzed by Raman. It is OK as standard method. However, it is better to check IR or XPS to investigate possible presence of the other functional groups.
4) In Figure 6, spectra in dashed lines are not well represented. Use of solid line with different colors is recommended.
5) References are rather old, Addition of recent papers to reference is recommended. Especially, recent rather papers in general papers on graphene, GO, and related materials can be added more (for example, see https://www.journal.csj.jp/doi/abs/10.1246/bcsj.20210297, https://pubs.rsc.org/en/content/articlelanding/2022/RA/D2RA00611A, https://link.springer.com/article/10.1007/s40820-021-00624-4, https://www.journal.csj.jp/doi/abs/10.1246/bcsj.20190368).
6) Figure 15 is important. However, the mechanisms are represented only simply. More details of formation mechanisms have to be included in the figure.
7) The presented work would have important contributions to applications and industry. Please add more detailed future perspectives in Conclusion section.
Reviewer 2 Report
The paper presents a possible interesting investigation concerning a sensitivity experimental activity for recycling GO from CFRP, through an electrochemical method.
However:
the work presents purely empirical results; the theoretical criterion for the design of the specimens, the percolation optimization, the electrical model should be outlined to highlight a scientific rigor.
The materials themselves are not so novel, so the methodological approach should be prevalent. Improvements in sensor gauge factor values are reported but without a strong analytical support or conceptual phase. The style needs to be revised (the first person is better to be avoided).
The study should be further enhanced to be considered in a journal as this one. At the moment, it seems much more a Technical Report.
In view of the above considerations, the reviewer doesn’t believe the manuscript ready for publication on a journal as Nanomaterials.
Round 2
Reviewer 1 Report
Replies and revisions are fine. The revised version becomes acceptable.
Author Response
Thanks for the kind reply.
Reviewer 2 Report
Dear authors, thanks for the revision.
Best regards.
Author Response
Thanks for your kind reply.